# Non-adherence to long-lasting insecticide treated bednet use following successful malaria control in Tororo, Uganda

**John Rek[1], Alex Musiime[1], Maato Zedi[1], Geoffrey Otto[1], Patrick Kyagamba[1], Jackson Asiimwe Rwatooro[1], Emmanuel Arinaitwe[1,2], Joaniter Nankabirwa[1,3], Sarah G. Staedke[2], Chris Drakeley[2], Philip J. Rosenthal[4], Moses Kamya[1,3], Grant Dorsey[4], Paul J. Krezanoski[4] ***

**1** Infectious Disease Research Collaboration, Kampala, Uganda, **2** London School of Hygiene and Tropical Medicine, London, United Kingdom, **3** Makerere University College of Health Sciences, Kampala, Uganda, **4** University of California, San Francisco, CA, United States of America

* paul.krezanoski@ucsf.edu

**Data Availability Statement:** Data from the cohort study is available through a novel open-access clinical epidemiology database resource, ClinEpiDB. Data (referred to as "PRISM2") can be

## Abstract

Indoor residual spraying (IRS) and long-lasting insecticide-treated bednets (LLINs) are common tools for reducing malaria transmission. We studied a cohort in Uganda with universal access to LLINs after 5 years of sustained IRS to explore LLIN adherence when malaria transmission has been greatly reduced. Eighty households and 526 individuals in Nagongera, Uganda were followed from October 2017 –October 2019. Every two weeks, mosquitoes were collected from sleeping rooms and LLIN adherence the prior night assessed. Episodes of malaria were diagnosed using passive surveillance. Risk factors for LLIN non-adherence were evaluated using multi-level mixed logistic regression. An age-matched case-control design was used to measure the association between LLIN non-adherence and malaria. Across all time periods, and particularly in the last 6 months, non-adherence was higher among both children <5 years (OR 3.31, 95% CI: 2.30–4.75; p<0.001) and school-aged children 5–17 years (OR 6.88, 95% CI: 5.01–9.45; p<0.001) compared to adults. In the first 18 months, collection of fewer mosquitoes was associated with non-adherence (OR 3.25, 95% CI: 2.92–3.63; p<0.001), and, in the last 6 months, residents of poorer households were less adherent (OR 5.1, 95% CI: 1.17–22.2; p = 0.03). Any reported non-adherence over the prior two months was associated with a 15-fold increase in the odds of having malaria (OR 15.0, 95% CI: 1.95 to 114.9; p = 0.009). Knowledge about LLIN use was high, and the most frequently reported barriers to use included heat and low perceived risk of malaria. Children, particularly school-aged, participants exposed to fewer mosquitoes, and those from poorer households, were less likely to use LLINs. Non-adherence to LLINs was associated with an increased risk of malaria. Strategies, such as behavior change communications, should be prioritized to ensure consistent LLIN use even when malaria transmission has been greatly reduced.

found at https://clinepidb.org/ce/app/record/dataset/DS_51b40fe2e2.

**Funding:** Funding was provided by the National Institutes of Health as part of the International Centers of Excellence in Malaria Research (ICMER) program (U19AI089674) and supported by a National Institute of Allergy and Infectious Disease Career Mentored Award (PJK; K23AI139364). JIN is supported by the Fogarty International Center (Emerging Global Leader Award grant number K43TW010365). EA is supported by the Fogarty International Center of the National Institutes of Health under Award Number D43TW010526. The funders had no role in the study design, data collection and analysis, decision to publish or preparation of the manuscript.

**Competing interests:** The authors have declared that no competing interests exist.

## Introduction

Long lasting insecticide-treated bednets (LLINs) are a mainstay of malaria prevention [1]. LLINs provide both individual protection from mosquitoes and a broader community effect via vector control. Despite increasingly robust programs for the universal distribution of LLINs, the impressive declines in malaria transmission achieved from 2000 to 2015 have recently stalled and may be reversing, especially in high burden African countries [2]. This stall in progress in malaria reduction has highlighted the importance of other tools for vector control such as indoor residual spraying of insecticides (IRS).

Evidence as to whether the addition of IRS to LLINs has a beneficial effect on reducing transmission is mixed. A recent comprehensive review of randomized trials concluded that caution was warranted, since some studies showed an additive effect, but other studies did not [3]. For example, a study in Gambia reported no significant difference in the density of indoor biting vectors caught in light traps in households receiving both LLINs and IRS compared to households receiving LLINs alone [4]. Similar findings have been reported in Benin [5]. A sub-group analysis from the comprehensive review, however, demonstrated that in populations with high LLIN adherence (reported use > 50%), adding IRS was associated with a substantial reduction in parasite prevalence, with a risk ratio of 0.47 (95% confidence interval [CI]: 0.33 to 0.67) compared to populations without IRS. Thus, the benefits of combining IRS with LLINs may depend on the levels of LLIN adherence.

The decision to use an LLIN, when one is available, is up to the individual or their parent/guardian. This is in contrast to decisions about IRS, which are in the hands of government officials as long as people consent to spraying of their households. There are a variety of well-described barriers to LLIN ownership and use that affect individual decision making, such as knowledge of malaria transmission, perceptions of risk and a sense of individual agency [6,7]. In addition, it is well documented that LLIN use changes in response to environmental factors such as seasonal rainfall and changes in temperature [8–10]. Less well described is how LLIN use changes over time and what factors are most determinative of individual decisions to use LLINs when transmission of malaria begins to decline in settings of intense malaria control. Identifying these dynamics in LLIN use and the mechanisms through which they act are crucial for policy makers to design effective LLIN promotion strategies to sustain malaria control once achieved.

Given that LLIN adherence may play a central role in the effectiveness of IRS and that successful IRS campaigns may have an effect on subsequent LLIN behaviors, we designed this study to explore LLIN non-adherence in a cohort living in an area of Eastern Uganda with historically high transmission intensity where transmission was dramatically reduced following two rounds of universal LLIN distribution and over five years of sustained IRS implementation [11]. Longitudinal measures of mosquito exposure, LLIN use and malaria episodes were used to 1) describe how LLIN use changed over time, 2) identify household, entomological and individual characteristics associated with non-adherence to LLINs, 3) estimate how non-adherence affected mosquito biting and the odds of being diagnosed with malaria, and 4) describe perceptions of malaria risk among participants.

## Materials and methods

### Study setting

The study took place in Nagongera sub-county in the Tororo District of Eastern Uganda. As part of two national campaigns, universal distribution of free LLINs was conducted in the district in November 2013 and repeated in May 2017. IRS with the carbamate bendiocarb was

first initiated in December 2014 –January 2015, with additional rounds administered in June-July 2015 and November-December 2015. In June-July 2016, the formulation of insecticide used for IRS was changed to the organophosphate pirimiphos-methyl (Actellic), with repeated rounds in June-July 2017, June- July 2018, and March-April 2019. These malaria interventions coincided with a significant reduction in malaria transmission, from an estimated entomological inoculation rate (EIR) of 238 infective bites per person per year prior to IRS being implemented, to an EIR of only 0.43 after 5 years of IRS [14].

## Study design and participants

Full details on enrollment and follow-up procedures for the cohort used in this study have been described previously [14]. In brief, 80 households and all of their inhabitants were recruited in October 2017 and followed through October 2019. The cohort was dynamic such that over the course of the study any permanent residents that joined the household were enrolled and residents leaving the household were withdrawn. All 80 enrolled households remained enrolled until the end of the cohort. Four hundred and sixty-six (466) participants were enrolled initially, with 65 individuals added (either born into or establishing residency in a cohort household) and 62 participants either dying or moving away, resulting in a total of 469 participants at the study end. At enrollment, household characteristics were assessed for the creation of a wealth index, as described elsewhere [12]. All designated sleeping rooms and sleeping areas were mapped and enumerated. All cohort households were provided additional LLINs at enrollment to ensure coverage of all sleeping spaces, and LLINs were provided on demand during monthly clinic visits for any households needing a new or replacement LLIN. Cohort study participants were encouraged to come to a dedicated study clinic open 7 days per week for all their medical care. As part of the broader cohort activities, routine visits were scheduled every 4 weeks for clinical assessments, malaria surveillance and measurement of other malaria risk factors, including a standardized evaluation for any overnight travel outside of the sub-county. Study participants found to have a tympanic temperature > 38.0˚C or history of fever in the previous 24 hours at the time of any clinic visit had a thick blood smear read urgently. If the thick smear was positive, the patient was diagnosed with malaria and managed according to national guidelines. Study subjects who missed their scheduled routine visits were visited at home and requested to come to the study clinic as soon as possible.

## Data collection

**Entomological surveillance.** Mosquito collections were conducted every 2 weeks in all cohort study households. In each room where cohort study participants slept, a miniature CDC light trap (Model 512; John W. Hock Company, Gainesville, Florida, USA) was positioned 1 m above the floor. Traps were set at 7 PM and collected at 7 AM the following morning. Female *Anopheles* mosquitoes were subsequently identified taxonomically and dissected, with each mosquito classified as either blood fed, not blood fed or unable to assess (for example due to damage in processing).

**LLIN adherence measures.** Every two weeks, on the morning after the CDC light traps were collected, a structured questionnaire was administered to an adult respondent in each household to gather information about reported LLIN use for each study participant the prior night, and where the participant slept the previous night (for assigning individual mosquito exposure) (S1 File).

**Exit interviews.** In November-December 2019, at the conclusion of the study, a semi-structured questionnaire was administered to the remaining enrolled participants (469) to inquire about their perceived risk of malaria, knowledge about malaria transmission,

community norms in relation to LLIN use, and indications for and barriers to LLIN use. Children under 12 years were aided by their parents in responding to this survey based on their perceptions and habits. Attempts were made to administer this exit interview to all cohort participants with multiple home visits, but, since the main cohort study had come to a close, logistical barriers resulted in only 459 of the total 469 participants being located for this final questionnaire (S2 File).

## Study endpoints

**Factors associated with non-adherence.**  For the analysis of risk factors for non-adherence, LLIN use for each individual was defined as reported use or non-use the prior night. The following factors were identified as potentially associated with LLIN non-adherence and included in the model: age and gender of the individual, household wealth index (in tertiles) and the total number of anopheles mosquitoes captured from the room where the individual slept during the night for which LLIN use was reported. Based on effect modification by calendar time on associations between our risk factors of interest and LLIN non-adherence, the analysis was stratified into two time periods: November 2017 through April 2019 (18 months) and May 2019 through October 2019 (6 months). Based on the distribution of the data, plausible categorization of differences in LLIN use behaviors and for aid in interpretation, age was stratified into three categories: under 5 years, 5 to 17 years and $\geq$ 18 years. A household wealth index was generated based on ownership of various assets using principal components analysis and LLIN non-adherence in the poorest households (lowest tertile) was compared to the least poor households (all other households). Finally, based on the distribution of the data and association with the outcome, the total number of female *Anopheles* mosquitoes captured from the room where an individual slept was categorized as either 0 to 2 versus 3 or more.

**Association between LLIN non-adherence and number of captured mosquitoes having taken a blood meal.**  Mosquito bites are an intermediate link between LLIN non-adherence and acquiring malaria. We hypothesized that the number of mosquitoes captured from participant rooms classified as blood fed, i.e. representing a potentially infectious bite, would be higher during a night when residents in the room reported non-adherence to LLINs. We chose to use blood fed mosquitoes as a marker of potential infection, and not the more traditional sporozoite rate, because we identified only nine mosquitoes with sporozoites out of a total of 15,780 collected. Such a low number would not have provided adequate power to support our inquiry. By room and date, we generated a variable representing the number of total blood fed mosquitoes, after adjusting for mosquitoes unable to be assessed for blood fed status due to damage in processing. We classified LLIN adherence, by room and date, as either complete LLIN adherence by all participants or any non-adherence reported by a room participant. Finally, since we reasoned that more people would represent both a higher potential lure for meal seeking mosquitoes and more available targets for biting, we created a variable accounting for the number of people sleeping in the room.

**Case-control design for associations between LLIN non-adherence and risk of malaria episodes.**  A case-control design was employed which identified age-matched controls for each case of malaria based on the date of diagnosis for each episode of malaria included in the analysis. A total of 38 cases of malaria were diagnosed over the 2 year follow-up period. No participant had more than one case of malaria. Malaria cases were excluded when prior LLIN use could not be assessed (n = 6), persistent asymptomatic parasitemia preceded the diagnosis of malaria (n = 6), and travel outside of the district was reported in the prior month (n = 4). These exclusions resulted in 22 cases of malaria included in the analyses. All age-matched controls available on the date malaria was diagnosed were included. Whenever possible, controls

were matched based on the year of age of the case. For three of the cases among older individuals (17, 29 and 38 years of age respectively), this constraint was relaxed to +/- 2 years in order to identify an adequate number of controls for each case. A mean of 22 age-matched controls (range 10 to 35) were identified per case. The main exposure of interest was individual LLIN use reported at the biweekly household visits and aggregated over various time windows. To account for a minimum incubation period of 7 days from an infectious mosquito bite to the onset of clinical symptoms of malaria, time windows for assessment of LLIN adherence were defined as 1–5 weeks, 1–9 weeks and 1–13 weeks prior to the date when a case/control was identified. The mean number of mosquitoes captured from the participant's room during the biweekly CDC light trap collections was included as a covariate and calculated similarly to LLIN use for the corresponding time windows.

## Statistical analysis

LOWESS smoothing was used to visually display trends in LLIN adherence and vector density over time. For the analysis of risk factors associated with LLIN non-adherence, a multi-level mixed effects logistic regression model was fit to account for both the multiple measures at the household and individual levels and the clustering of participants within the same households. In assessing the association between blood fed mosquitoes and LLIN non-adherence, we utilized a multi-level mixed effects negative binomial regression model, accounting for multiple measures at the room level, the hierarchical clustering of rooms within households and adjusting for the number of participants in the room. The measure of association is reported as a risk ratio with 95% confidence intervals (CIs) comparing the prevalence of blood fed mosquitoes, of the total captured, in rooms with and without any reported LLIN non-adherence. For the case-control analysis, conditional logistic regression models were fit for each time window of LLIN non-adherence exposure and adjusted for mean mosquito exposure. Measures of associations for this analysis and the risk factors associated with LLIN adherence were reported as odds ratios (ORs) with 95% CIs. All statistical analyses were conducted at the 5% significance level and were performed using STATA 14 (StataCorp LP. 2015. College Station, TX).

## Ethics approval and consent to participate

Written informed consent for participation in the cohort study was obtained in the appropriate language from the adults and children. In addition, children above 8 years also provided assent for study participation. For children under 8 years, consent was provided by the parents or guardians. Additional verbal consent was obtained prior to the exit interview questionnaire. Study ethical approval was obtained from Uganda National Council for Science and Technology (UNCST), Makerere University School of Medicine Research Ethics Committee, University of California, San Francisco Committee for Human Research and the London School of Hygiene and Tropical Medicine Ethics Committee.

## Results

### Characteristics of study participants

A total of 80 households were enrolled, with 526 individuals providing data on LLIN adherence. There was an average of 5.8 individuals per household (SD: 1.3), 1.8 rooms per household used for sleeping (SD: 0.6) and 1.7 sleeping spaces per room (SD: 0.7). At enrollment, 98% of households owned at least one LLIN, with an average of 3.5 (SD: 1.6) LLINs per household. The number of LLINs was not measured after enrollment, but was assumed to be adequate to cover all sleeping areas due to on demand distribution to cohort participants. Fifty-

two percent of participants were female and the average age at enrollment was 15.9 years (SD: 16.1; range: 1 month to 76 years). At enrollment, 34% (177/526) of participants were under five years of age, 38% (201/526) were 5–17 years of age, and 28% (148/526) were 18 years or older.

## Changes in mosquito density, malaria episodes and LLIN use over time

Over the two year study period, a total of 15,780 female *Anopheles* mosquitoes were collected during the biweekly collections, resulting in an average density of 2.1 mosquitoes per room per night. Ninety-nine percent of mosquitoes collected were identified as *Anopheles arabiensis* by PCR and one percent were *Anopheles gambiae sensu stricto*. Each year there was a large peak in mosquito density following the long rainy season, which occurs from April-June, and a smaller peak following the short rainy season, which occurs from October-November. Following the short rainy seasons, there were periods when almost no mosquitoes were collected. This period extended from December-April during the second year, possibly because the last round of IRS was administered early, in March 2019. There were a total of 38 malaria cases diagnosed over the 2 year follow-up period. Small clusters of cases were evident following the two large peaks in mosquito density (**Fig 1**).

During the first year of follow-up, mean LLIN adherence fluctuated from 50–85%, generally decreasing during periods of lower mosquito density and increasing during periods of higher mosquito density. From November 2018 to February 2019, mean LLIN adherence decreased precipitously, falling below 10%. This marked decline corresponded with an extended 6-month period when almost no female *Anopheles* mosquitoes were collected. During the last 6 months of follow-up, mean LLIN adherence increased, reaching over 40%, and corresponding with a sharp rise in mosquito density (**Fig 1**).

Since LLINs are used differently by different age groups, changes in mean reported LLIN adherence over time were stratified by age categories (i.e. under 5 years, 5 to 17 years and ≥18 years). The three age groups had a qualitatively similar pattern over time, but, over the entire study period, LLIN adherence was consistently higher among adults and lowest among school-aged children (5 to 17 years). Reported adherence among children under five years of

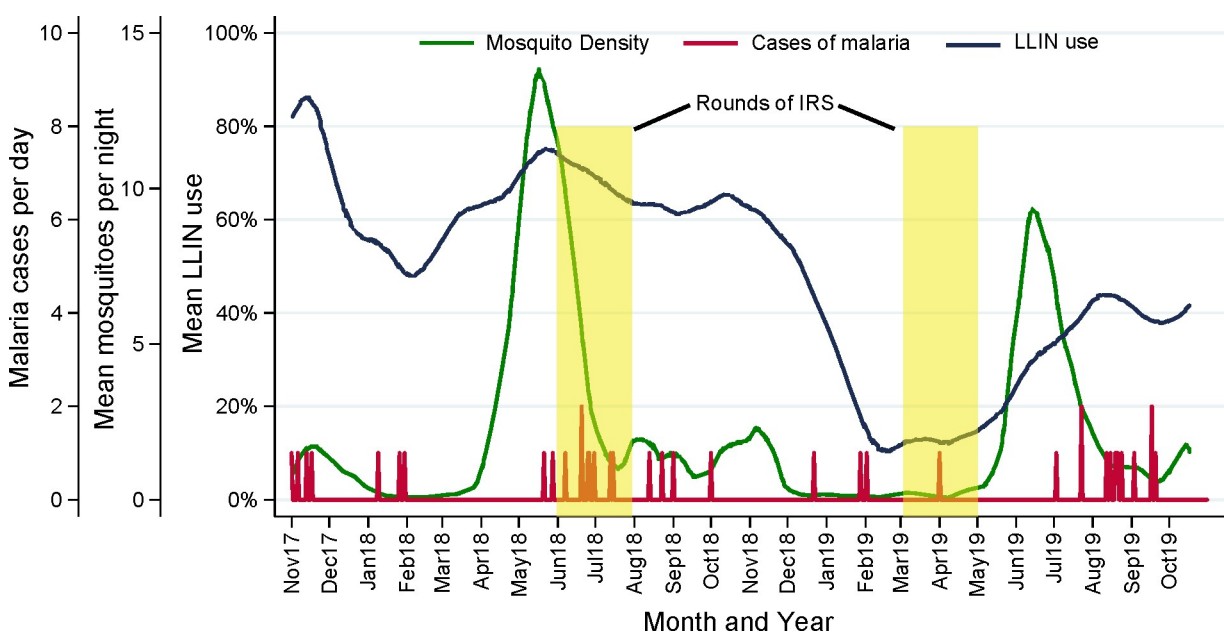

**Fig 1. Changes in LLIN use, mosquito counts and malaria cases over time.**

age was higher than among school age children, but persistently less than that for adults across the entire study. All three groups had a similar nadir in LLIN adherence in February 2019, but as LLIN use began to rise in May 2019, there is evidence that adults were more likely to use LLINs than children, reaching a peak of 65% use, compared to 45% in children under 5 years, and only 30% in children 5 to 17 years of age (**Fig 2**).

## Factors associated with LLIN non-adherence

During the first 18 months of follow-up, age was significantly associated with the odds of reported LLIN non-adherence after adjusting for mosquito exposure, household wealth and gender. Compared to adults, both children under 5 years and those aged 5 to 17 years were less likely to be adherent: OR 1.47 (95% confidence interval (CI): 1.27 to 1.70; p<0.001) and 2.67 (95% CI: 2.34 to 3.06; p<0.001), respectively. In this time period, individuals sleeping in rooms where 0–2 mosquitoes were captured per night had 3.25 (95% CI: 2.92 to 3.63; p<0.001) the odds of non-adherence compared to those sleeping in rooms where 3 or more mosquitoes were captured per night. There was no association between household wealth and non-adherence during the first 18 months of follow-up (**Table 1**). In the final six months, differences in the odds of LLIN non-adherence according to age increased, such that, compared to adults, children under 5 years had 3.31 times the odds (95% CI: 2.30 to 4.75; p<0.001) and children 5 to 17 years had 6.88 times the odds (95% CI: 5.01 to 9.45; p<0.001) of LLIN non-adherence. There was no association between vector density and non-adherence in the final six months of the study, but individuals living in the poorest tertile of households had 5.09 (95% CI: 1.17 to 22.2; p = 0.03) the odds of non-adherence compared those living in wealthier households. There was no association between gender and non-adherence in either the first 18 months or final six months of follow-up.

## Association between LLIN non-adherence and blood-fed mosquitoes

The prevalence of blood fed mosquitoes was 6.3% (95% CI: 5.3% to 7.4%) in rooms where everyone reported LLIN adherence compared to 12.0% (95% CI: 10.5% to 12.4%) in rooms

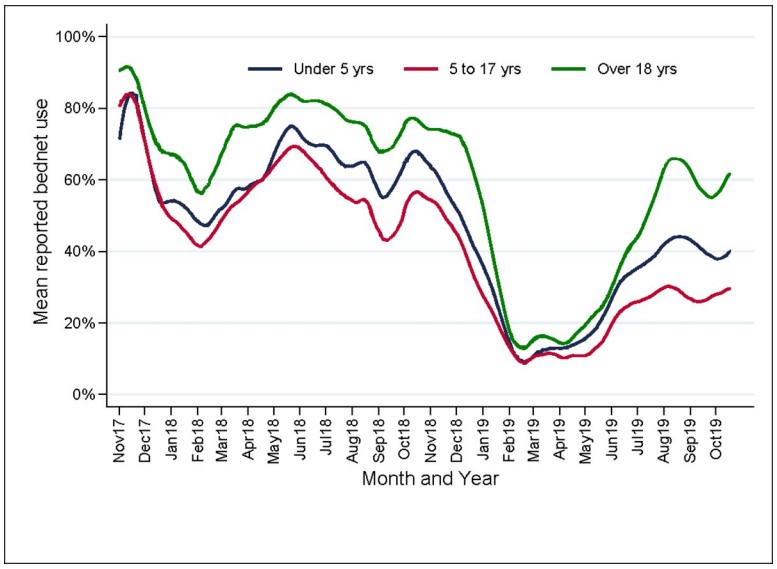

**Fig 2. Changes in LLIN use over time stratified by age category.**

**Table 1. Risk factors for LLIN non-adherence stratified by calendar time.**

| Risk factor | Categories | November 2017 –April 2019 (18 months) | | | | May 2019 –October 2019 (6 months) | | | |
|---|---|---|---|---|---|---|---|---|---|
| | | Observations | Non-adherence | OR (95% CI) | p-value | Observations | Non-adherence | OR (95% CI) | p-value |
| Age in years | ≥ 18 | 5155 | 37.7% | reference group | | 1668 | 51.6% | reference group | |
| | < 5 | 4772 | 47.6% | 1.47 (1.27–1.70) | <0.001 | 1263 | 65.6% | 3.31 (2.30–4.75) | <0.001 |
| | 5 - <18 | 7358 | 54.6% | 2.67 (2.34–3.06) | <0.001 | 2568 | 75.5% | 6.88 (5.01–9.45) | <0.001 |
| Vector density [a] | ≥ 3 | 2711 | 32.1% | reference group | | 1433 | 64.5% | reference group | |
| | 0–2 | 14574 | 50.5% | 3.25 (2.92–3.63) | <0.001 | 4066 | 66.5% | 0.99 (0.80–1.23) | 0.91 |
| Household wealth [b] | Least poor | 11017 | 47.5% | reference group | | 3509 | 60.0% | reference group | |
| | Poorest | 6268 | 47.8% | 1.03 (0.54–1.97) | 0.94 | 1990 | 76.5% | 5.09 (1.17–22.2) | 0.03 |
| Gender | Male | 8344 | 48.7% | reference group | | 2583 | 66.6% | reference group | |
| | Female | 2879 | 46.6% | 0.98 (0.88–1.10) | 0.76 | 2916 | 65.4% | 1.15 (0.87–1.52) | 0.34 |

[a] Number of female anopheles captured using CDC light traps the prior night in the room the participant was sleeping.

[b] Wealth index stratified into the poorest households (lowest tertile) and the least poor households (all other households).

with any reported non-adherence (p<0.001). In the mixed effects model, adjusting for multiple measures and the number of people sleeping in the room, any reported LLIN non-adherence was associated with 2.2 times (95% CI: 1.8 to 2.7; p>0.001) the prevalence of blood fed mosquitoes. Interestingly, after adjusting for LLIN adherence, the number of participants in the room showed no linear trend in increasing prevalence of blood fed mosquitoes captured (p = 0.99).

## Associations between prior LLIN non-adherence and malaria episodes

The results of the case-control study show that any reported LLIN non-adherence was associated with increased odds of being diagnosed with malaria after adjusting for mosquito exposure. The strength of this association increased when non-adherence was assessed over longer windows of time preceding the date malaria was diagnosed (Table 2). Compared to those who always reported LLIN use, the odds of being diagnosed with malaria was 3 times higher (OR 3.12, 95% CI: 1.06–9.21, p = 0.04) in individuals who reported not using their LLIN at least once over a one month period (from 1 to 5 weeks prior to the date of diagnosis), and 15 times higher (OR 15.0, 95% CI: 1.95–114.9, p = 0.009) among those who reported not using their LLIN at least once over a 2 month period. When adherence was assessed over a three month period, the odds of being diagnosed with malaria could not be estimated because none of the cases (0/22) reported full adherence, whereas 26.4% (121/458) of controls were fully adherent at all measurements.

**Table 2. Association between LLIN non-adherence and odds of malaria episode.**

| Period of assessment[a] | Reported LLIN adherence[b] | Proportion among cases (%) | Proportion among controls (%) | OR (95% CI)[c] | p-value |
|---|---|---|---|---|---|
| 1 to 5 weeks prior | At each assessment | 5/22 (22.7%) | 196/458 (42.8%) | reference group | |
| | Less than always | 17/22 (77.3%) | 262/458 (57.2%) | 3.12 (1.06–9.21) | 0.04 |
| 1 to 9 weeks prior | At each assessment | 1/22 (4.6%) | 163/458 (35.6%) | reference group | |
| | Less than always | 21/22 (95.5%) | 295/458 (64.4%) | 15.0 (1.95–114.9) | 0.009 |
| 1 to 13 weeks prior | At each assessment | 0/22 (0%) | 121/458 (26.4%) | reference group | |
| | Less than always | 22/22 (100%) | 337/458 (73.6%) | Unable to estimate | N/A |

[a] Relative to date of diagnosis of case (or comparable control).

[b] Measured every 2 weeks.

[c] Controlling for mean vector density during period of assessment from room where participant slept.

## Knowledge and perceptions of malaria risk in the cohort

As noted above, after multiple home visits after the end of the cohort study in an attempt to reach all cohort participants, only 459 of the total 469 participants were administered the LLIN adherence questionnaire. When interviewed at the end of the study, 93.7% (430/459) of individuals reported they thought that the malaria risk had decreased in the community compared to the prior year. Knowledge of malaria transmission was heterogeneous, with 93.2% (428/459) of individuals identifying mosquitoes as a potential cause of malaria, but only 31.8% (146/459) of participants reporting that only mosquitoes can transmit malaria. Despite reported LLIN adherence being lower in children than adults, knowledge of priority groups for LLIN use was high; 96.1% of participants identified children under 5 years of age, 94.1% pregnant women and 80.1% school-aged children as individuals who should use an LLIN every night (Table 3). Many individuals knew how to use an LLIN correctly (88.7%). Reasons that an individual might not use an LLIN that is hung above a sleeping space, included 'too hot' (85.6%),

**Table 3. Knowledge and perceptions of malaria risk relating to LLIN adherence.**

| Question | Options | Observations | Proportion |
|---|---|---|---|
| Which of these groups should use a bednet every night?[a] | Children <5 years | 441/459 | 96.1% |
| | Pregnant women | 433/459 | 94.3% |
| | School age children | 371/459 | 80.1% |
| | Adults | 365/459 | 79.5% |
| Do you feel that you know how to use a bednet correctly? | Yes | 407/459 | 88.7% |
| | No | 50/459 | 10.9% |
| | Unsure | 2/459 | 0.4% |
| What are some important reasons why someone would not use a bednet that is hung above their sleeping space?[a] | Too hot | 393/459 | 85.6% |
| | No Mosquitoes | 132/459 | 28.8% |
| | Bed bugs/fleas | 113/459 | 24.6% |
| | Forgot | 76/459 | 15.6% |
| | Don't like smell | 53/459 | 11.5% |
| | Net too dirty | 44/459 | 9.6% |
| | Itching, rashes, allergic reaction | 36/459 | 7.8% |
| | Net too old/too many holes | 30/459 | 6.5% |
| | No malaria | 29/459 | 6.3% |
| | Don't know how to use | 15/459 | 3.2% |
| | Net no longer kill insects | 12/459 | 2.6% |
| | Inconvenient to use | 6/459 | 1.3% |
| | Others (spread infection; fire risk) | 9/459 | 2.0% |
| | Unsure | 8/459 | 1.7% |
| Please bring to mind the last night you recall not using a bednet. Can you tell me why you did not use a bednet during that night? | I just forgot | 162/459 | 35.3% |
| | I was travelling | 85/459 | 18.5% |
| | It was too hot | 77/459 | 16.8% |
| | Not applicable, I never miss a night | 74/459 | 16.1% |
| | There were no mosquitoes around | 34/459 | 7.4% |
| | There is no malaria here | 13/459 | 2.8% |
| | Not sure | 14/459 | 3.1% |

[a] Multiple responses accepted.

no mosquitoes around (28.8%) or no malaria (6.3%). Interestingly, 24.6% of individuals reported the presence of bedbugs/fleas as a potential barrier to individual LLIN use. When asked about recent experience with LLIN non-adherence, participants reported that they simply forgot (35.2%), were travelling (18.5%), it was too hot (16.8%), there were no mosquitoes (7.4%), and that there was no malaria (2.8%).

## Discussion

In this cohort of households from a district in Uganda experiencing more than a 500-fold reduction in malaria transmission following universal LLIN distribution and 5 years of sustained IRS, we identified various important findings in relation to how LLIN use has changed over time. First, a marked decline in individual-reported LLIN use was observed: only 20% of individuals reporting LLIN use from February to June 2019, compared to an average of 60% during the same time period in the prior year. LLIN non-adherence was significantly higher in children compared to adults, and highest in school-aged children, despite widespread reported knowledge of the importance of ensuring nightly LLIN use by children. The finding of poor LLIN use among school age children has been well-described elsewhere [13], but the consistently low adherence among children under five years of age was surprising. In addition to age, other factors identified as associated with lower LLIN adherence included lower household wealth and the presence of fewer mosquitoes in the room in which an individual slept. Any reported LLIN non-adherence was associated with 2.2 times the prevalence of mosquitoes captured that had taken a blood meal, establishing an intermediate link between poor LLIN adherence and increased risk of potentially infectious mosquito bites. Finally, using an age-matched case control design, not using an LLIN was associated with increased odds of having a malaria episode, confirming the importance of using LLINs even with very low levels of malaria transmission.

The steep decline in reported LLIN use in 2019 was remarkable given that all study participants had access to LLINs and that malaria transmission had not changed much compared to the prior year. Whether a drop of this magnitude in LLIN use is unusual is not clear, as most data on changes in LLIN use come from repeated cross-sectional studies rather than the prospective cohort design that we employed [14,15]. A cohort study in Kenya reported how LLIN use impacted mortality over time, but did not report serial measures of LLIN use [16]. Other studies in the context of free universal distribution campaigns have also noted sharp declines in reported use over time, but have often attributed this to a decline in ownership of LLINs [17]. Changes in ownership should not have significantly affected our cohort, since access to an LLIN was universal. Anecdotal reports from health officials in the district noted an outbreak of bedbugs leading people to use their LLINs less, and 25% of cohort participants reported bedbugs as a potential reason for not using an LLIN. Associations between household insect infestations, IRS and LLIN use have been reported previously, but the evidence is mixed on the direction of associations with LLIN adherence, as concerns regarding bedbugs have been identified as both facilitators [18] and barriers [19] to LLIN use.

The use of LLINs during successful IRS campaigns is important, as there is evidence of an interaction between LLINs and IRS [20], and additive effects could depend on levels of LLIN adherence [3]. Additionally, when IRS campaigns have been stopped, marked resurgence in the burden of malaria has been described despite universal LLIN distribution [21]. There is interest in promoting LLIN use at the time of stopping IRS, and in introducing next generation LLINs with novel synergists or insecticides, as an "exit strategy" [22]. In this study, it appears that perceptions of risk of malaria transmission decreased as community burden and mosquito densities declined. Along with this decrease in actual and perceptions of risk, it is possible that

other salient barriers to use, such as the heat (referenced by 85.6% of respondents), bedbugs (24.6%), not liking the smell (11.5%) or the bednet being too dirty (9.6%), overcome other incentives to consistently use LLINs and lead to lower adherence.

Even when malaria transmission, and overall mosquito populations are significantly decreased [14], by a combined strategy of IRS, universal access to LLINs and case management with artemisinin-based combination therapies, LLIN adherence still has an important effect on an individual's risk of malaria. In this cohort, even when malaria transmission was reduced to historically low levels, not using an LLIN at every biweekly measurement over two months was associated with a 15-fold increase in the odds of being diagnosed with malaria, after controlling for age and adjusting for mosquito density. Thus, maintaining high and consistent LLIN use should remain an important priority for individual protection from malaria, especially for high risk groups.

Of particular interest in this cohort was the association between age and LLIN non-adherence. Not surprisingly, school age children (age 5 to 17) were consistently poor users of LLINs, as has been demonstrated elsewhere [8,23,24]. In addition, though nearly everyone in the cohort understood that children under five should use an LLIN every night, adults used LLINs significantly more than children. In the final six months of the study, this difference was multiplied two-fold. This is particular interesting given that it might be expected that children under five will often use their bednets under the supervision of adults in the household. The discordance between LLIN use by adults and young children deserves special attention in future efforts to understand who is responsible for choices related to bednet use in Ugandan households. Given their heightened risk of severe malaria, the lower use of LLINs by young children in this study highlights the necessity of more robust social and behavior change campaigns that can reach beyond knowledge and address actual behaviors.

Another finding of interest was that individuals from the poorest households were less likely to use an LLIN in the last six months of the study, when LLIN use was generally lower. The effect of wealth on LLIN use is mixed and may be context specific. While most studies show that wealth is inversely associated with LLIN use [25,26], other studies have not shown this effect [10]. One plausible mechanism for decreased LLIN use among poorer households, and one that may have implications for IRS settings, could be LLIN attrition. As malaria risk declines, the value of LLINs to the owner may decline as well, incentivizing the sale or trade of the LLIN. This seems unlikely in this study, however, as the households had free access to LLINs.

This study had important limitations. Extending our conclusions to other populations is limited by the fact that the households were enrolled in a specialized cohort, with access to free medical care, access to free LLINs on demand and frequent household visits, which may have affected the incidence of malaria, knowledge and behaviors compared to the general population. For the questionnaire related to LLIN adherence, due to logistical constraints at the end of the cohort study, only 459 of the total 469 participants were administered the exit interview. This could have introduced bias, although multiple home visits were attempted and there is no reason to suspect that the households who were administered the survey would differ systematically from the overall cohort in respect to their perceptions about LLINs. The use of LLINs was reported every two weeks by one adult in the household for all other members of the household. This may have introduced biases in relation to reported LLIN use, but longitudinal measures of use are likely more reliable than one time cross sectional measures, as used in standard Malaria Indicator Surveys and most other methods of assessing LLIN use. Social desirability bias may also have been a factor, though it might be expected to have inflated reported use, whereas in this study LLIN use was reported to be quite low. In particular, social desirability bias does not seem plausible as the driver of low relative LLIN use for children, because presumably it would be socially undesirable to report non-use in children.

## Conclusion

As malaria transmission declined in our study setting with sustained IRS, LLIN use declined as well. Despite understanding the importance of LLIN adherence among children, adults were more likely to use LLINs than children and non-adherence was strongly associated with a higher odds of malaria. Future research should focus on the dynamic interplay between LLIN adherence and changing perceptions of malaria risk. Social and behavior change communication programs and other strategies for sustaining high levels of LLIN use should be prioritized to ensure consistent LLIN adherence, even in the setting of successful malaria control and reduced transmission.

## Supporting information

**S1 File. Biweekly household questionnaire.**
(DOC)

**S2 File. Exit interview instrument.** Exit interview questionnaire relating to bednet adherence.
(DOCX)

## Acknowledgments

We thank the study team and the Infectious Diseases Research Collaboration (IDRC) for administrative and technical support. We are grateful to the study participants who participated in this study and their families.

## Author Contributions

**Conceptualization:** John Rek, Chris Drakeley, Philip J. Rosenthal, Grant Dorsey, Paul J. Krezanoski.

**Formal analysis:** John Rek, Alex Musiime, Grant Dorsey, Paul J. Krezanoski.

**Investigation:** Alex Musiime, Maato Zedi, Geoffrey Otto, Patrick Kyagamba, Jackson Asiimwe Rwatooro, Emmanuel Arinaitwe, Joaniter Nankabirwa, Grant Dorsey, Paul J. Krezanoski.

**Methodology:** Alex Musiime, Paul J. Krezanoski.

**Project administration:** John Rek, Maato Zedi, Geoffrey Otto, Patrick Kyagamba, Jackson Asiimwe Rwatooro, Emmanuel Arinaitwe, Joaniter Nankabirwa, Sarah G. Staedke, Paul J. Krezanoski.

**Supervision:** John Rek, Emmanuel Arinaitwe, Joaniter Nankabirwa, Sarah G. Staedke, Chris Drakeley, Philip J. Rosenthal, Moses Kamya, Grant Dorsey, Paul J. Krezanoski.

**Writing – original draft:** John Rek, Grant Dorsey, Paul J. Krezanoski.

**Writing – review & editing:** John Rek, Alex Musiime, Maato Zedi, Geoffrey Otto, Patrick Kyagamba, Jackson Asiimwe Rwatooro, Emmanuel Arinaitwe, Joaniter Nankabirwa, Sarah G. Staedke, Chris Drakeley, Philip J. Rosenthal, Moses Kamya, Grant Dorsey, Paul J. Krezanoski.

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
