## [Decision Letter · Decision Letter 0]

24 Sep 2020

PONE-D-20-27114

Non-adherence to long-lasting insecticide treated bednet use following successful malaria control in Tororo, Uganda

PLOS ONE

Dear Dr.  Krezanoski,

Thank you for submitting your manuscript for review to PLoS ONE. After careful consideration, we feel that your manuscript will likely be suitable for publication if the authors revise it to address critical points raised by the reviewers.  According to reviewers, there are  specific areas where further improvements would be of substantial benefit to the readers. According to reviewer # 1 a number of methodological issues need to be addressed by the authors, otherwise it could potentially affect the transparency, accuracy and reliability of the findings (details can be found in the reviewer’s comments). Reviewer #2 also suggests clarifying the study design, including the criteria used to classify household.  For your guidance, a copy of the reviewers' comments was included below.

We look forward to receiving your revised manuscript.

Kind regards,

Luzia Helena Carvalho, Ph.D.

Academic Editor

PLOS ONE

Journal Requirements:

Reviewers' comments:

Reviewer's Responses to Questions

**Comments to the Author**

1. Is the manuscript technically sound, and do the data support the conclusions?

Reviewer #1: No

Reviewer #2: Yes

2. Has the statistical analysis been performed appropriately and rigorously? 

Reviewer #1: Yes

Reviewer #2: Yes

3. Have the authors made all data underlying the findings in their manuscript fully available?

Reviewer #1: Yes

Reviewer #2: No

4. Is the manuscript presented in an intelligible fashion and written in standard English?

Reviewer #1: Yes

Reviewer #2: Yes

5. Review Comments to the Author

Reviewer #1: General

This is a longitudinal cohort study aimed to explore non-adherence of long-lasting insecticide treated bed-net (LLIN) use by measuring mosquito exposure, and malaria episodes in a malaria high transmission area which the cases significantly reduced after LLIN distribution and IRS implementation. It is an interesting study and quite appreciative for the authors conducting a comprehensive study related to behavior, perceived risk of malaria transmission, entomology and history of illness in this study. However, there are several significant methodological problems that need to be discusses carefully, which could potentially affect the transparency, accuracy and reliability of the findings (details can be found in the comments below).

Introduction

The introduction well written. The aim of this study clearly stated. However, authors need to further review existing literature and provide examples that relevant with association between the use of LLIN and number of captured mosquitoes. Authors could add and cite factors association between the non-adherence of LLIN and risk of malaria transmission papers and provide more explanation or justification on the novelty and rationale of this study – to make this introduction stronger and convincing and explain how their study will provide additional insight to the effectiveness LLIN implementation program.

What is the significance or expected benefit of the findings of this study? Please add.

Method

The authors are suggested to re-organized the sections and would be good if they could provide flowchart illustrating steps of analyses. There is critical point in the methods – the authors need to carefully describe how to choose respondent for the exit interviews as it was mentioned cohort enrollment was dynamic.

Specific comments:

• Authors could provide flowchart illustrating cohort participants enrollment including for the exit interviews.

• Please describe activities on routine visits, why did scheduled every 4 weeks? Please explained.

• It is good to present the methods by heading and sub-headings. Study setting, Study design and participants, data collection with sub-headings entomological survey, risk factors associated with LLIN non-adherence, exit interviews, and then move to statistical analysis.

• It should be stated the species of mosquitoes captured in the study based on vector or suspected vector malaria

• Why was age stratified into three categories: under 5 years, 5 to 17 years and ≥ 18 years? Age range should be considered when determining adherence. it should be used adult age range.

• Authors may cite / add literature to categorize the number of persons sleeping within LLIN (0-2 Vs 3 or more)

• Why LLIN non-adherence was measured on the basis of the number of mosquitoes caught having taken a blood meal instead of those positive for sporozoites? Please explain

Result

• Authors should compile the section of headings more concise and systematic. For example: characteristics of the study participants, factors associated between non-adherence of LLINs use, mosquito biting and malaria episodes; predictors of non-adherence of LLINs Use

• Line 310: Authors should be opening by explaining the total sample outcome for the exit interview, since the number for the cohort was different from the total sample, please add.

Discussion

It would be better if the authors would opening/introductory paragraph by describing the main result of study.

Conclusions

Authors should be more specific to address regarding changing perception of malaria risk.

Reviewer #2: Review of the paper by Rek et al Non-adherence to long-lasting insecticide treated bednet use following successful malaria control in Tororo, Uganda

Non adherence to bed net could reduce the effectiveness of LLINs. My comments on the manuscript are as follows

Method

The author say the classify household as poor or not poor but they didn’t explain which criteria they used to assess the household wealth this information need to be added.

Line 121 – 122 In addition to EIR data, also add information on anopheline species in the study area (composition and density)

Line 148 to 152 « Every two weeks, on the morning after the CDC light traps were collected, a structured questionnaire was administered to an adult respondent in each household to gather information …» Please add the questionnaire as an additional file

Line 154 to 158 « Exit interviews

In November-December 2019, at the conclusion of the study, a semi-structured questionnaire was administered to each participant inquiring about perceived risk of malaria, knowledge about malaria transmission, community norms in relation to LLIN use, and indications for and barriers to LLIN use. » please add questionnaire as an additional file

Line 188 - 192 « A total of 38 cases of malaria were diagnosed over the 2 year follow-up period. Malaria cases were excluded when prior LLIN use could not be assessed (n=6), persistent asymptomatic parasitemia preceded the diagnosis of malaria (n=6), and travel outside of the district was reported in the prior month (n=4). These exclusions resulted in 22 cases of malaria included in the analyses. » Did the authors registered people having two or more malaria attacks during the study period and how did they proceeded with the analysis

Page 11 line 258 « Since LLIN behaviors vary by age, changes in mean reported LLIN adherence … » Please what is « LLIN behaviour » ?

Results

The authors should mention anopheline species collected

The authors need to say whether they had cases of people who drop out from the study and give the attrition rate.

Table 1

« N » is placed for what ? please explain

First line Age in years

The age range « 5 - <18 » (is it 18 years) Also the author used the sign « < » does this means that children of 1 day old were included in the study?

Table 2

The first line 4/22 and 17/22 does not sum to 22/22 please verify your calculations

Table 3

First line N refer to what ?

Discussion

The fact that non adherence to LLINs use was associated to more blood fed mosquitoes indoor and increase risk of malaria transmission does it mean no effect of IRS on mosquito populations ? and what could be the influence of insecticide resistance can the authors discuss this further in the discussion section.

6. PLOS authors have the option to publish the peer review history of their article (what does this mean?). If published, this will include your full peer review and any attached files.

Reviewer #1: No

Reviewer #2: No

---

## [Author Response · Author response to Decision Letter 0]

11 Oct 2020

“Non-adherence to long-lasting insecticide treated bednet use following successful malaria control in Tororo, Uganda”.

Response to Reviewers:

Journal Requirements

As requested, we have made adjustments based on the referenced style templates. In particular, we have removed the key words, funding, word count and author contributions from the title page. We have also retitled the main sections to reflect the template (i.e. Background is now Introduction). We have also indented all paragraphs and reformatted the abstract to be unstructured.

Thank you for this request, which we note was also the subject of Reviewer 1’s comments. We have provided additional information in the Materials and Methods section (page 6) in regards to the procedures for identifying respondents to the exit survey, explanations for the discrepancy in numbers as was a concern of Reviewer 1 (459 exit interview respondents versus 469 remaining cohort participants) and clarification of the role of parents in aiding their children in responding to the questions. We have also clarified in the ethics approval section (page 9) that additional verbal consent was obtained from cohort participants for this questionnaire. Finally, we have added the verbal consent script and exit interview questionnaire relating to bednet adherence that was administered to the cohort study participants as a supplementary file (S2). 

Reviewer Comments

1. “…authors need to further review existing literature and provide examples that relevant with association between the use of LLIN and number of captured mosquitoes.”

Thank you for this suggestion. As requested, we have added two additional citations that highlight the uncertain additive effects of IRS in addition to LLINs on vector density measured, as in our study, with CDC light traps. We have added the following text on page 3:

“For example, a study in Gambia reported no significant difference in the density of indoor biting vectors caught in light traps in households receiving both LLINs and IRS compared to households receiving LLINs alone [4]. Similar findings have been reported in Benin [5].”

2. “Authors could add and cite factors association between the non-adherence of LLIN and risk of malaria transmission papers…”

To bolster our discussion of additional factors associated with non-adherence to LLINs, we have added an additional sentence and two more citations that highlight some of the other well-described barriers to LLIN use (page 3; line 60):

“There are a variety of well-described barriers to LLIN ownership and use that affect individual decision making, such as knowledge of malaria transmission, perceptions of risk and a sense of individual agency [9,10].”

3. “…[P]rovide more explanation or justification on the novelty and rationale of this study…[w]hat is the significance or expected benefit of the findings of this study? Please add.”

Thank you for pointing out that we could do better at highlighting the added value from this study. We have edited the sentence on the top of page 4 to read:

“Less well described is how LLIN use changes over time and what factors are most determinative of individual decisions to use LLINs when transmission of malaria begins to decline in settings of intense malaria control. Identifying these dynamics in LLIN use and the mechanisms through which they act are crucial for policy makers to design effective LLIN promotion strategies to sustain malaria control once achieved.” 

4. “There is critical point in the methods – the authors need to carefully describe how to choose respondent for the exit interviews as it was mentioned cohort enrollment was dynamic.”

We appreciate this concern and have added in the methods section a more complete description of the procedures for identifying the subjects for the exit interviews (page 6; line 122):

“…a semi-structured questionnaire was administered to the remaining enrolled participants to inquire about their perceived risk of malaria, knowledge about malaria transmission, community norms in relation to LLIN use, and indications for and barriers to LLIN use. Children under 12 years were aided by their parents in responding to this survey based on their perceptions and habits. Attempts were made to administer this exit interview to all cohort participants with multiple home visits, but, since the main cohort study had come to a close, logistical barriers resulted in only 459 of the total 469 participants being located for this final questionnaire.”

We have additionally added to the limitations section (page 19; line 371)a discussion of any potential bias this may represent:

“For the questionnaire related to LLIN adherence, due to logistical constraints at the end of the cohort study, only 459 of the total 469 participants were administered the exit interview. This could have introduced bias, although multiple home visits were attempted and there is no reason to suspect that the households who were administered the survey would differ systematically from the overall cohort in respect to their perceptions about LLINs.”

5. “Authors could provide flowchart illustrating cohort participants enrollment including for the exit interviews.”

We appreciate that the full study flowchart is useful for reference. Given that this study was based on a subset of the total cohort activities, we opted to refer to a recently published article in which the full details are described, including a flowchart illustrating enrollment and the dynamic nature of the cohort (see reference #14: Nankabirwa et al. Malaria Transmission, Infection, and Disease following Sustained Indoor Residual Spraying of Insecticide in Tororo, Uganda. Am J Trop Med Hyg. 2020 Jul 20. doi: 10.4269/ajtmh.20-0250). To make clear the flow of participants we added on page 5, line 97, a brief description of the dynamic cohort:

“All 80 enrolled households remained enrolled until the end of the cohort. Four hundred and sixty-six participants were enrolled initially, with 65 individuals added (either born into or establishing residency in a cohort household) and 62 participants either dying or moving away, resulting in a total of 469 participants at the study end.”

6. “Please describe activities on routine visits, why did scheduled every 4 weeks? Please explained.”

We have added additional information on activities at the routine visits and why they were scheduled every 4 weeks on page 5, line 101. We do not elaborate on activities that were performed outside the scope of the current study and instead reference to the full details of the cohort activities published elsewhere (see above):

“As part of the broader cohort activities, routine visits were scheduled every 4 weeks for clinical assessments, malaria surveillance and measurement of other malaria risk factors, including a standardized evaluation for any overnight travel outside of the sub-county.”

7. “It is good to present the methods by heading and sub-headings. Study setting, Study design and participants, data collection with sub-headings entomological survey, risk factors associated with LLIN non-adherence, exit interviews, and then move to statistical analysis.”

As requested, in the Materials and methods section we have retitled our headings and sub-headings to clearly separate study setting, study design, data collection, study endpoints and statistical analysis.

8. It should be stated the species of mosquitoes captured in the study based on vector or suspected vector malaria

Thank you for noting this important omission. We have added an additional line on page 10, line 218: 

“Ninety-nine percent of mosquitoes collected were identified as Anopheles arabiensis by PCR and one percent were Anopheles gambiae sensu stricto.”

9. “Why was age stratified into three categories: under 5 years, 5 to 17 years and ≥ 18 years? Age range should be considered when determining adherence. it should be used adult age range.”

On reviewing the data and identifying plausible categories for age ranges, we felt that these three categories capture the data best and map well with other studies of age determinants of LLIN use. Our categories include the less than 5 year old, pre-school children, the five to 17 year old, “school-age children” and adults who are 18 years old and older. These seem to us to allow the best interpretation of how activities with age-associations track with LLIN use behaviors.

10. “Authors may cite / add literature to categorize the number of persons sleeping within LLIN (0-2 Vs 3 or more)”

Thank you for identifying this important point. We typically think of the number of individuals using an LLIN as a measure of availability of nets when supply is limited. The nature of our cohort study, where all households had free access on demand to LLINs at any time to cover all sleeping areas, made this distinction less important in our minds. For that reason, we did not measure the number of individuals under each net, and thus were not able to categorize then as suggested. We believe that since we are assuming that all sleeping areas have bednets and we have individualized reports of LLIN use, the number of individuals under each LLIN is less important to our overall outcomes such as blood-fed mosquitoes and malaria episodes.

11. “Why LLIN non-adherence was measured on the basis of the number of mosquitoes caught having taken a blood meal instead of those positive for sporozoites? Please explain”

Thank you for this question. Unfortunately, due to the extremely low levels of malaria transmission in our cohort, we did not identify adequate numbers of sporozoite-positive mosquitoes to support our analysis. We added on page 7, line 155:

“We chose to use blood fed mosquitoes as a marker of potential infection, and not the more traditional sporozoite rate, because we identified only nine mosquitoes with sporozoites out of a total of 15,780 collected. Such a low number would not have provided adequate power to support our inquiry.”

12. “Authors should compile the section of headings more concise and systematic. For example: characteristics of the study participants, factors associated between non-adherence of LLINs use, mosquito biting and malaria episodes; predictors of non-adherence of LLINs Use”

Thank you for this helpful suggestion. We have changed some of the sub-headings to make them more clear. For example we changed “Demographics” to “Characteristics of study participants” (page 10, line 209) and “Association between LLIN non-adherence and blood-fed mosquitoes” (page 12; line 265).

13. “Line 310: Authors should be opening by explaining the total sample outcome for the exit interview, since the number for the cohort was different from the total sample, please add.”

This is another helpful suggestion. Besides the additional information on the exit interview procedure that we have added to the methods and limitations sections, already discussed above, we have also added in this section a reminder of why the total number administered the questionnaire differ from the total in the cohort (page 15, line 289):

“As noted above, after multiple home visits after the end of the cohort study in an attempt to reach all cohort participants, only 459 of the total 469 participants were administered the LLIN adherence questionnaire.”

14. “It would be better if the authors would opening/introductory paragraph by describing the main result of study.”

Thank you for requesting this clarification. We have added a phrase highlighting the main findings of the study in the first paragraph of the discussion (page 15, line 305):

“In this cohort of households from a district in Uganda experiencing more than a 500-fold reduction in malaria transmission following universal LLIN distribution and 5 years of sustained IRS, we identified various important findings in relation to how LLIN use has changed over time. First, a marked decline in individual-reported LLIN use was observed: only 20% of individuals reporting LLIN use from February to June 2019, compared to an average of 60% during the same time period in the prior year.”

15. “Authors should be more specific to address regarding changing perception of malaria risk.”

Thank you for this suggestion. However, since the exit interviews were only administered once at the end of the cohort study, we are unable to make clear conclusions about how perceptions of malaria risk may have changed over time. We can hypothesize that the reason for the decrease in use of LLINs may be a result of changes in perceptions, but we unfortunately are unable to make that conclusion based off only one measurement at the end of the cohort. Future work would ideally measure perceptions of risk at baseline and hope to capture changes in those perceptions in relation to the level of malaria transmission.

Reviewer 2

1. “The author say the classify household as poor or not poor but they didn’t explain which criteria they used to assess the household wealth this information need to be added.”

Thank you for identifying this omission. We have included a reference to a recent paper from our group which explains the components and procedures utilized to create the wealth index used for this study.

Page 5 line 98:

“At enrollment, household characteristics were assessed for the creation of a wealth index, as described elsewhere [15].”

2. “Line 121 – 122 In addition to EIR data, also add information on anopheline species in the study area (composition and density)”

Thank you for this suggestion. As also requested above, we have added information on the species of anophelines that were captured in the study on page 10, line 222:

“Ninety-nine percent of mosquitoes collected were identified as Anopheles arabiensis by PCR and one percent were Anopheles gambiae sensu stricto.”

3. “Line 148 to 152 « Every two weeks, on the morning after the CDC light traps were collected, a structured questionnaire was administered to an adult respondent in each household to gather information …» Please add the questionnaire as an additional file”

Thank you for this request. We have added the biweekly questionnaire as a supplementary file 1. 

4. Line 154 to 158 « Exit interviews. In November-December 2019, at the conclusion of the study, a semi-structured questionnaire was administered to each participant inquiring about perceived risk of malaria, knowledge about malaria transmission, community norms in relation to LLIN use, and indications for and barriers to LLIN use. » please add questionnaire as an additional file”

We have added the exit interview questionnaire as supplementary file 2.

5. “Did the authors registered people having two or more malaria attacks during the study period and how did they proceeded with the analysis”

Thank you for this question. In fact, no participant had more than one malaria episode during the study. We have added an additional sentence to clarify this on page 8, line 163:

“No participant had more than one case of malaria.”

6. Page 11 line 258 « Since LLIN behaviors vary by age, changes in mean reported LLIN adherence … » Please what is « LLIN behaviour » ?

We have clarified that we mean LLINs are used differently by different age groups. We have altered the sentence to now say:

“Since LLINs are used differently by different age groups…”

7. “The authors need to say whether they had cases of people who drop out from the study and give the attrition rate.”

As above, we reference the publication with the full study details and also add on page 5, line 97 a brief description of the attrition rate:

 “All 80 enrolled households remained enrolled until the end of the cohort. Four hundred and sixty-six participants (466) were enrolled initially, with 65 individuals added (either born into or establishing residency in a cohort household) and 62 participants either dying or moving away, resulting in a total of 469 participants at the study end.”

8. “Table 1 « N » is placed for what ? please explain”

We have adjusted the “N” to indicate “Observations” in Table 1, page 14.

9. “First line Age in years The age range « 5 - <18 » (is it 18 years) Also the author used the sign « < » does this means that children of 1 day old were included in the study?”

The first (5 to <18) indicates from 5 years up to just under 18 years of age, but not 18 years. Yes, the less than 5 year age group includes just born children. We think this is correct in the inequalities as written.

10. “Table 2. The first line 4/22 and 17/22 does not sum to 22/22 please verify your calculations”

Thank you for this close attention. There was a typographical error and the first should be 5/22 (=22.7%). We have adjusted this in Table 2 on page 14.

11. “Table 3. First line N refer to what ?”

To make this clearer, we have adjusted the “N” to indicate “Observations” in Table 3, page 16.

12. “The fact that non adherence to LLINs use was associated to more blood fed mosquitoes indoor and increase risk of malaria transmission does it mean no effect of IRS on mosquito populations?” 

Thank you for this question. We have highlighted in the discussion that even with low mosquito populations and low malaria incidence there is still a significant impact of LLIN use. Despite historically low values in both of these metrics, LLIN use is still of value in individual protection from mosquito bites and malaria episodes.

“Even when malaria transmission, and overall mosquito populations are significantly decreased [14], by a combined strategy of IRS, universal access to LLINs and case management with artemisinin-based combination therapies, LLIN adherence still has an important effect on an individual’s risk of malaria.”

13. “what could be the influence of insecticide resistance can the authors discuss this further in the discussion section.”

Thank you for this question. This is obviously a very important consideration. However, we do not feel that this paper’s focus on LLIN use allows us to make conclusions about insecticide resistance, either to IRS nor to LLINs. Our findings indicate that even with very low community transmission and lower than historical mosquito densities, both findings which would not be expected if there was widespread and significant insecticide resistance, LLIN use still provides value for individual protection. Thus, we avoid making conclusions about insecticide resistance and instead focus our attention on the importance of understanding changes in LLIN adherence and how policymakers might perhaps address low LLIN adherence in particular groups where it still matters.

---

## [Decision Letter · Decision Letter 1]

26 Oct 2020

PONE-D-20-27114R1

Non-adherence to long-lasting insecticide treated bednet use following successful malaria control in Tororo, Uganda

PLOS ONE

Dear Dr. Krezanoski,

After careful consideration, we feel that your manuscript will likely be suitable for publication if the authors revise it to address critical points raised by the reviewer.  According to reviewer, there are some specific areas where further improvements would be of substantial benefit to the readers.   For your guidance, a copy of the reviewers' comments was included below. 

We look forward to receiving your revised manuscript.

Kind regards,

Luzia Helena Carvalho, Ph.D.

Academic Editor

PLOS ONE

Reviewers' comments:

Reviewer's Responses to Questions

**Comments to the Author**

1. If the authors have adequately addressed your comments raised in a previous round of review and you feel that this manuscript is now acceptable for publication, you may indicate that here to bypass the “Comments to the Author” section, enter your conflict of interest statement in the “Confidential to Editor” section, and submit your "Accept" recommendation.

Reviewer #1: All comments have been addressed

Reviewer #2: All comments have been addressed

2. Is the manuscript technically sound, and do the data support the conclusions?

Reviewer #1: Yes

Reviewer #2: Yes

3. Has the statistical analysis been performed appropriately and rigorously? 

Reviewer #1: Yes

Reviewer #2: Yes

4. Have the authors made all data underlying the findings in their manuscript fully available?

Reviewer #1: Yes

Reviewer #2: Yes

5. Is the manuscript presented in an intelligible fashion and written in standard English?

Reviewer #1: Yes

Reviewer #2: Yes

6. Review Comments to the Author

Reviewer #1: I appreciate the effort and response of the authors to my previous comments and suggestions. However, before considering the manuscript for publication, there are some questions that need be discussed. My notes below on the revised manuscript are:

• line 221: It should be Result section, please changed

• line 235: Authors could discuss more rigorously regarding the findings "general compliance with LLIN decreases during periods of lower mosquito density and increases during periods of higher mosquito density" in line with the findings (line 303): the reason one might not use LLIN is 28.8 % answered that there were no mosquitoes compared to 'too hot' (85.6%). Please explain… considering understanding the importance of LLIN compliance is strongly associated with a higher likelihood of malaria

• Line 318: Author should add in the discussion section regarding non-adherence the used of LLINs in the age group under 5 years old. Considering the age range is still under the supervision of parents or adults, therefore I was clarified on my previous review note why was age range under 5 years old, included.

Reviewer #2: (No Response)

7. PLOS authors have the option to publish the peer review history of their article (what does this mean?). If published, this will include your full peer review and any attached files.

Reviewer #1: **Yes: **Mara Ipa

Reviewer #2: No

---

## [Author Response · Author response to Decision Letter 1]

10 Nov 2020

1) Line 221: It should be Result section, please change

As requested, we have corrected this typo and removed “and discussion” from line 221.

2) Line 235: Authors could discuss more rigorously regarding the findings "general compliance with LLIN decreases during periods of lower mosquito density and increases during periods of higher mosquito density" in line with the findings (line 303): the reason one might not use LLIN is 28.8 % answered that there were no mosquitoes compared to 'too hot' (85.6%). Please explain… considering understanding the importance of LLIN compliance is strongly associated with a higher likelihood of malaria

Thank you for highlighting this important behavioral point that highlights individuals’ reported reasons for the observed behaviors. We have edited line 235 to read: 

“In this study, it appears that perceptions of risk of malaria transmission decreased as community burden and mosquito densities declined. Along with this decrease in actual and perceptions of risk, it is possible that other salient barriers to use, such as the heat (referenced by 85.6% of respondents), bedbugs (24.6%), not liking the smell (11.5%) or the bednet being too dirty (9.6%), overcome other incentives to consistently use LLINs and lead to lower adherence.”

3) Line 318: Author should add in the discussion section regarding non-adherence the used of LLINs in the age group under 5 years old. Considering the age range is still under the supervision of parents or adults, therefore I was clarified on my previous review note why was age range under 5 years old, included.

This is another important point and we agree that more discussion of the intrahousehold behaviors related to age and bednet use is warranted. We have added the following text on line 318:

“This is particular interesting given that it might be expected that children under five will often use their bednets under the supervision of adults in the household. The discordance between LLIN use by adults and young children deserves special attention in future efforts to understand who is responsible for choices related to bednet use in Ugandan households.”

---

## [Decision Letter · Decision Letter 2]

19 Nov 2020

Non-adherence to long-lasting insecticide treated bednet use following successful malaria control in Tororo, Uganda

PONE-D-20-27114R2

Dear Dr.  Krezanoski,

We’re pleased to inform you that your manuscript has been judged scientifically suitable for publication and will be formally accepted for publication once it meets all outstanding technical requirements.

Kind regards,

Luzia Helena Carvalho, Ph.D.

Academic Editor

PLOS ONE

Additional Editor Comments (optional):

Reviewers' comments:

Reviewer's Responses to Questions

**Comments to the Author**

1. If the authors have adequately addressed your comments raised in a previous round of review and you feel that this manuscript is now acceptable for publication, you may indicate that here to bypass the “Comments to the Author” section, enter your conflict of interest statement in the “Confidential to Editor” section, and submit your "Accept" recommendation.

Reviewer #1: All comments have been addressed

Reviewer #2: All comments have been addressed

2. Is the manuscript technically sound, and do the data support the conclusions?

Reviewer #1: Yes

Reviewer #2: Yes

3. Has the statistical analysis been performed appropriately and rigorously? 

Reviewer #1: Yes

Reviewer #2: Yes

4. Have the authors made all data underlying the findings in their manuscript fully available?

Reviewer #1: Yes

Reviewer #2: Yes

5. Is the manuscript presented in an intelligible fashion and written in standard English?

Reviewer #1: Yes

Reviewer #2: Yes

6. Review Comments to the Author

Reviewer #1: (No Response)

Reviewer #2: I have no further comment to the authors. Comments have been adequately addressed. The paper looks good.

7. PLOS authors have the option to publish the peer review history of their article (what does this mean?). If published, this will include your full peer review and any attached files.

Reviewer #1: No

Reviewer #2: No

---

## [Editor Report · Acceptance letter]

24 Nov 2020

PONE-D-20-27114R2 

Non-adherence to long-lasting insecticide treated bednet use following successful malaria control in Tororo, Uganda 

Dear Dr. Krezanoski:

I'm pleased to inform you that your manuscript has been deemed suitable for publication in PLOS ONE. Congratulations! Your manuscript is now with our production department. 

Kind regards, 

on behalf of

Dr. Luzia Helena Carvalho 

Academic Editor

PLOS ONE